# Applying an Optimum Bucking Method to Comparing the Volume and Value Recovery of Cut-to-Length and Tree-Length Merchandizing Systems in Piedmont and the Coastal Plain

Carlos Diniz [1,*], Mathew Smidt [1] and John Sessions [2]

1    Southern Research Station, USDA Forest Service, Auburn, AL 36849, USA; mathew.smidt@usda.gov
2    College of Forestry, Oregon State University, Corvallis, OR 97331, USA; john.sessions@oregonstate.edu
*    Correspondence: carlos.diniz@usda.gov

**Abstract:** Dynamic prices and markets create value for contractors who can readily evaluate the gross and net income differences for alternative merchandizing systems. The majority of the southern U.S.A. relies on tree-length merchandizing, with occasional identification and merchandizing of logs for a specific market or specific tree dimensions or qualities. Cut-to-length (CTL) merchandizing has generated more value when compared to tree-length (TL) marketing, but these comparisons have been limited to specific stands and markets (specifications and prices). The study objective was to demonstrate a process for evaluating cut-to-length and tree-length merchandizing systems in their production of gross value by applying a dynamic programming stem-level optimum bucking approach that maximizes the stem value given specific market conditions. TL merchandizing resulted in a better volume recovery for both regions, but the value recovery was better for CTL merchandizing. Observing the value recovery by diameter class, DSH classes of up to 100 mm had a similar value in both merchandizing systems, but CTL merchandizing yielded a greater or similar value per cubic meter across the range for larger tree sizes. Access to tree data and merchandizing tools needs to be addressed so wood suppliers and landowners may benefit from stem optimization and sensor technology being embedded into modern harvesters and processors.

**Keywords:** bucking; dynamic programming; logging; optimization; loblolly pine



## 1. Introduction

Intensive forest management has increased forest productivity dramatically in the United States of America. The Pacific Northwest was once the leading region for timber harvest volume, but now the Southeast accounts for 63 percent of the annual harvest volume [1–3]. Timber production migrated from the Pacific Northwest to the US South as a result of federal policies such as the Northwest Forest Plan [4]. Georgia loggers harvest more than 70,000 tons per year, while loggers in the Coastal Plain of Virginia harvest 37,000 tons per year [4–6]. For [4], there are some attributes that explain the high production rates of southern loggers, such as gentle terrain, less seasonal downtime, and millions of hectares of pine plantations.

The deployment of harvesting systems and technology is dependent on many factors (e.g., economic, cultural). While harvesting systems in the southern U.S.A. appear to be uniform and somewhat static [4], over time, they have undergone relatively rapid changes. Both the weekly production and contractor number for shortwood systems decreased by over 70% from 1979 to 1987, while tree-length systems doubled in their weekly production [7]. At the time, the tree-length systems provided an increase in mechanization and a reduction in logging costs compared to shortwood systems [8]. The contemporary innovations between 1979 and 1987 attempted in shortwood production could not offset the in-wood and milling advantages of tree-length systems delivering tree-length logs [7].

Modern CTL equipment was viewed as a method to increase mechanization and improve merchandizing [9]. Regional studies in the late 1990s found a reduced gap in productivity between tree-length and cut-to-length harvesting systems [10]. When bucking is needed, the optimization routines and stem measurement available using modern harvesters and processor systems can increase yield and value [11–14]. As assortments are increased, processor heads may help operators maintain productivity [15]. Additionally, the bucking optimization of tree-length logs in mills allowed them control over the distribution and quality of lumber outputs [16].

Timber is delivered to mills based on the large- and small-end diameters specified by the receiving mills for the tree-length dimensions [4,17]. Different from cut-to-length system, bucking using a full-tree system can easily lead to errors since it relies on visual estimation of lengths, diameters, and defects on the part of the knuckleboom operator [14,18]. Bucking in forests using a harvester head has the potential for an increased volume and value recovery and increases in mill efficiency resulting from receiving logs of a standard length [19].

Traditional bucking may not guarantee the optimum bucking solution because the trees are cut from the logger's perspective [20]. Maximizing the value recovery by bucking the stems is defined as optimum bucking [21]. Optimal bucking has been studied by several researchers [21–24] and can be categorized into three levels: first, solving stem-level problems seeking to maximize the total stem value. Second, solving stand-level problems to determine the maximum aggregate production value. Third, solving forest-level problems to maximize the global profit considering demand constraints, merchandizing, restrictions, and the forest state [25].

Optimum bucking problems can be solved using modern optimization methods such as network analysis, linear programming, dynamic programming, and heuristic techniques [26,27]. Sessions et al. developed an optimum bucking solution called BUCK [28]. The program was able to increase the timber volume and value by up to 14%. Garland et al. also improved the timber volume and value (22%) using computer-aided bucking at the stump [29]. Wang et al. studied stem-level optimal bucking and found an increase of 14% in the gross log value [30]. Linear programming and dynamic programming have been generally integrated to formulate optimum bucking problems [25,31]. Heuristic techniques such as genetic algorithms and tabu search also have been used in optimum bucking studies [26,32].

The introduction and adoption of alternative systems always face a variety of hurdles; however, a significant factor important to landowners, loggers, wood buyers, and mills is the system impact on value after in-wood merchandizing. The gross value following merchandizing is dependent on the number of products, the delivered value of the products, and the stem characteristics. Product differentiation and value may be affected by the wood basket or procurement area, as well as stem characteristics related to genetics, silviculture, and stand history. Since many factors are local, the objective of this analysis was to demonstrate a process for evaluating cut-to-length and tree-length merchandizing systems for the production of gross value by applying a dynamic programming stem-level optimum bucking approach that maximizes the stem value given specific market conditions.

## 2. Materials and Methods

### 2.1. Markets and Products

We narrowed down potential locations by using the list of 20 Southern Research Station Experimental Forests and randomly selected one location in the Piedmont region (Hitchiti) and another in the Coastal Plain (Santee). The market area was defined as a circle with a 125 km radius centered on a public road intersection at the edge of the forest. We contacted all mills which used pine within the radius and asked for the specifications (diameters and lengths) for all the products purchased at the location. For the pricing, we used the TimberMart-South mill prices from Q2 2019 from Region 1 in Georgia (Hitchiti) and Region 2 in South Carolina (Santee). No significant price changes were observed in these areas from 2018 to 2020.

From the specifications of the products used by the sawmills in each region, we identified all the possible bucking possibilities for a specific log. We applied the optimum bucking method to virtually simulate for each region which harvesting system would be able to achieve the best results in terms of the volume and value recovered. These specifications allowed us to verify the products that could be extracted from a particular tree. And since the goal is to maximize the recovered value, a specific combination will be the best. For example, from a tree, we can obtain one sawlog and two pulpwood logs, or alternatively, one sawlog and one chip-n-saw log. The chosen ones will be the ones with a higher recovery value.

### 2.2. Tree Data

We acquired tree data on 4957 loblolly and shortleaf pine trees from a stand in east central Mississippi cut and bucked using a modern cut-to-length harvester. We used an onboard computer (OBC) to capture and store the data in StanForD Classic formatted stem files (*.stm) [33,34]. Individual tree data were extracted from the data set using the StanForD2Tbl software from FP Innovations. These data files provided a stem number, diameter at stump height (DSH), and stem diameter every 10 cm to merchantable height. The tree sizes were distributed by their DSH classes for the sample distribution (Figure 1). Most (83%) of the trees with a stump diameter greater than 10 cm were bucked past a 10 cm top, and 70% were bucked past an 8 cm top. For all trees with a stump diameter of at least 25 cm, the average merchantable height to a 15 cm top was 10.9 m with a range from 9.8 to 14.8 m. For the bucking analysis, we randomly selected 57 trees from the dataset—3 per 2 cm stump diameter class. The volume of the tree sections was calculated using the Smalian equation for every 10 cm section. All the diameters, calculations, and specifications were applied outside of the bark. The desired log volume was obtained from the sum of these sections.

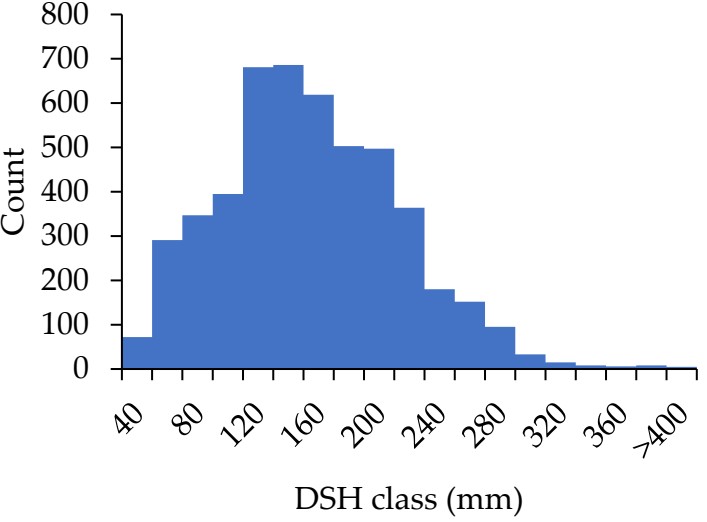

**Figure 1.** Frequency distribution by diameter at stump height (DSH) classes (20 mm) for all trees from the harvester.

After selecting the 57 trees and calculating the volume of each section, we identified all possible bucking combinations based on the specifications mentioned in the previous section, which represent each of the studied regions according to the harvesting system. With this, we organized a list of the 57 trees and their bucking points. This is a list with four columns: tree number (N), first bucking point (n1), last bucking point (n), and recovered value (i), which consists of multiplying the volume by the value of the assortment to which such log belongs.

*2.3. Bucking*

We identified the maximum gross value by applying an algorithm that recognizes a special network structure. For value maximization, stem-level optimal bucking [35] was used. The optimum bucking algorithm focuses on one line of code—an update to the temporarily best value of each node as each line of the sorted list is processed. The optimal bucking solution is found by finding the highest value path from node *1* at the base of the tree to node *n* at the top of the tree [23]. The network consists of exactly N comparisons where N equals the number of possible logs. The arcs are the bucking options. An arc is defined by its beginning node, ending node, and value. After the optimal value has been determined over the length of the tree, the predecessor node is used to retrace the optimal path from the end node to node 1 to identify the optimal bucking. In this problem, the log length is represented as an arc between the nodes, as shown in Figure 2.

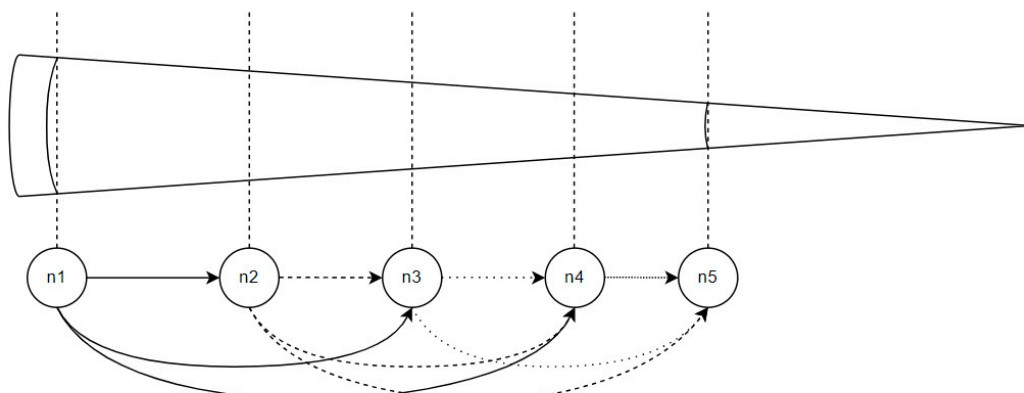

**Figure 2.** Network representation of a sample tree. The nodes (e.g., n1) represents possible bucking points, and the arrows represent the arcs or logs.

Encoding of the bucking algorithm using VBA to maximize the value recovery is described below (Algorithm 1):

---

**Algorithm 1:** Encoding of the bucking algorithm using VBA to maximize the value recovery

---

$DO\ i = 1; nArcs$

{

If bestvalue(begnode(i)) + value(i) > bestvalue(endnode(i))

{bestvalue(endnode(i)) = bestvalue(begnode(i)) + value(i)
prednode(endnode(i)) = begnode(i)}

}

where

nArcs = number of bucking options (arcs) over the length of the tree
begnode(i) = the start of an arc (log)
endnode(i) = the end of an arc (log)
bestvalue(i) = the current highest value at point (i)
value(i) = the log value from the begnode(i) to the endnode(i)
prednode(i) = the predecessor node used for tracing back the optimal path

---

After identifying the node with the highest value at the end node, the recursion technique is used from its predecessor node, as described below (Algorithm 2):

---

**Algorithm 2:** Algorithm to trace back over the optimal path to identify logs to be cut in the optimal solution

TOP = L
K = *prednode(L)*
DO WHILE K > 1

{

TOP = K
K = prednode(K)

}

LOOP

where

L = the merchantable length of the tree
prednode(L) = the arc that presents the highest value for the desired tree
prednode(K) = the predecessor arc used to execute the optimal bucking

## 3. Results

### 3.1. Markets and Products

We identified 15 Coastal Plain and 12 Piedmont area mills which accepted pine round-wood products and received product specifications from 9 and 6 mills, respectively. Similar specifications for pulpwood, chip-n-saw, and sawtimber were consolidated across mills, and the delivered prices were applied to each market specification, with the same price for CTL and TL products (Table 1).

**Table 1.** Product specifications, including length, minimum large end diameter (LED) and minimum small end diameter (SED), and delivered value in US$ per cubic meter (nominal 2019) for each region and system, or both.

| Region | System | Product | Length (m) | LED (cm) | SED (cm) | Value ($/m$^3$) |
|--------|--------|---------|-----------|----------|----------|-----------------|
| Piedmont | CTL | Sawtimber | 3.8 | 35 | 20 | 43.39 |
| | CTL | Sawtimber | 5.0 | 35 | 20 | 43.39 |
| | Both | Sawtimber | 8.2 | 35 | 15 | 43.39 |
| | TL | Sawtimber | >8.2 | 35 | 15 | 43.39 |
| | CTL | Chip-n-saw | 2.4 | 20 | 10 | 36.15 |
| | CTL | Chip-n-saw | 3.6 | 20 | 10 | 36.15 |
| | CTL | Chip-n-saw | 4.9 | 20 | 10 | 36.15 |
| | Both | Chip-n-saw | 6.1 | 20 | 10 | 36.15 |
| | TL | Chip-n-saw | >6.1 | 20 | 10 | 36.15 |
| | CTL | Pulpwood | 3.0 | 10 | 2.5 | 30.24 |
| | Both | Pulpwood | 6.0 | 10 | 2.5 | 30.24 |
| | TL | Pulpwood | >6.0 | 10 | 2.5 | 30.24 |
| Coastal Plain | CTL | Sawtimber | 3.8 | 35 | 20 | 43.99 |
| | CTL | Sawtimber | 5.0 | 35 | 20 | 43.99 |
| | Both | Sawtimber | 7.6 | 35 | 15 | 43.99 |
| | TL | Sawtimber | >7.6 | 35 | 15 | 43.99 |
| | CTL | Chip-n-saw | 4.6 | 20 | 10 | 36.35 |
| | CTL | Chip-n-saw | 6.1 | 20 | 10 | 36.35 |
| | Both | Chip-n-saw | 7.6 | 20 | 10 | 36.35 |
| | TL | Chip-n-saw | >7.6 | 20 | 10 | 36.35 |
| | CTL | Pulpwood | 4.3 | 10 | 3.0 | 32.40 |
| | Both | Pulpwood | 6.1 | 10 | 3.0 | 32.40 |
| | TL | Pulpwood | >6.1 | 10 | 3.0 | 32.40 |

### 3.2. Bucking

A summary of the optimal bucking results on volume and value recovery for the 57 sampled trees are in Table 2. TL merchandizing resulted in a better volume recovery for both regions, but the value recovery was better for CTL merchandizing. Since we restricted the pulpwood length in CTL merchandizing to 3 or 6 m, there was more unrecovered volume (waste) in small trees (DSH < 160). For the CTL and TL merchandizing, the total waste using the Piedmont specifications was 0.50 and 0.25 cubic meters, respectively. Using the Coastal Plain specification total, the amount of waste was similar for CTL and TL merchandizing at 0.50 and 0.15 cubic meters.

**Table 2.** Summary of volume (m$^3$) and value (US$) recovery mean and standard deviation (SD) by 10 mm diameter stump height class (DSH) using the optimum bucking method.

| Region | DSH (mm) | CTL | | | | TL | | | |
|---|---|---|---|---|---|---|---|---|---|
| | | Mean (m$^3$) | SD (m$^3$) | Mean ($) | SD ($) | Mean (m$^3$) | SD (m$^3$) | Mean ($) | SD ($) |
| Piedmont | 40 | 0.008 | 0.001 | 0.26 | 0.01 | 0.010 | 0.001 | 0.33 | 0.01 |
| | 60 | 0.014 | 0.002 | 0.47 | 0.07 | 0.017 | 0.002 | 0.55 | 0.05 |
| | 80 | 0.033 | 0.007 | 1.08 | 0.22 | 0.036 | 0.005 | 1.16 | 0.17 |
| | 100 | 0.037 | 0.006 | 1.20 | 0.18 | 0.042 | 0.007 | 1.36 | 0.24 |
| | 120 | 0.055 | 0.015 | 1.86 | 0.60 | 0.060 | 0.013 | 1.94 | 0.42 |
| | 140 | 0.107 | 0.034 | 3.81 | 1.22 | 0.111 | 0.036 | 3.86 | 1.37 |
| | 160 | 0.135 | 0.028 | 4.75 | 1.01 | 0.141 | 0.028 | 4.86 | 1.17 |
| | 180 | 0.177 | 0.009 | 6.35 | 0.45 | 0.179 | 0.010 | 6.51 | 0.37 |
| | 200 | 0.211 | 0.058 | 7.62 | 2.14 | 0.200 | 0.043 | 7.28 | 1.55 |
| | 220 | 0.270 | 0.031 | 10.3 | 2.19 | 0.268 | 0.039 | 9.73 | 1.40 |
| | 240 | 0.302 | 0.040 | 12.2 | 2.47 | 0.303 | 0.039 | 11.6 | 1.65 |
| | 260 | 0.328 | 0.141 | 13.8 | 5.65 | 0.344 | 0.132 | 12.8 | 5.81 |
| | 280 | 0.375 | 0.233 | 15.1 | 10.2 | 0.358 | 0.225 | 14.5 | 9.92 |
| | 300 | 0.478 | 0.060 | 20.6 | 3.33 | 0.480 | 0.112 | 21.1 | 4.91 |
| | 320 | 0.614 | 0.146 | 26.4 | 6.38 | 0.613 | 0.146 | 25.7 | 6.12 |
| | 340 | 0.764 | 0.173 | 33.1 | 7.93 | 0.757 | 0.168 | 33.0 | 7.91 |
| | 360 | 0.392 | 0.321 | 16.6 | 14.1 | 0.388 | 0.295 | 16.4 | 13.7 |
| | 380 | 0.677 | 0.209 | 29.5 | 9.03 | 0.690 | 0.166 | 29.1 | 9.49 |
| | >400 | 0.745 | 0.478 | 31.8 | 22.5 | 0.812 | 0.530 | 31.3 | 24.6 |
| Coastal Plain | 40 | 0.010 | 0.001 | 0.33 | 0.01 | 0.010 | 0.001 | 0.33 | 0.01 |
| | 60 | 0.015 | 0.003 | 0.49 | 0.11 | 0.017 | 0.002 | 0.55 | 0.05 |
| | 80 | 0.031 | 0.006 | 1.00 | 0.19 | 0.036 | 0.005 | 1.16 | 0.17 |
| | 100 | 0.035 | 0.009 | 1.12 | 0.28 | 0.042 | 0.007 | 1.36 | 0.24 |
| | 120 | 0.058 | 0.014 | 2.01 | 0.49 | 0.060 | 0.013 | 1.94 | 0.42 |
| | 140 | 0.109 | 0.036 | 3.86 | 1.27 | 0.111 | 0.036 | 3.82 | 1.34 |
| | 160 | 0.139 | 0.029 | 4.94 | 1.05 | 0.141 | 0.028 | 4.92 | 0.96 |
| | 180 | 0.178 | 0.013 | 6.46 | 0.48 | 0.181 | 0.007 | 6.52 | 0.34 |
| | 200 | 0.200 | 0.036 | 7.23 | 1.36 | 0.203 | 0.040 | 7.32 | 1.50 |
| | 220 | 0.268 | 0.027 | 10.30 | 2.05 | 0.275 | 0.033 | 10.4 | 2.38 |
| | 240 | 0.301 | 0.037 | 12.29 | 2.44 | 0.303 | 0.039 | 11.6 | 1.69 |
| | 260 | 0.333 | 0.146 | 14.15 | 5.96 | 0.344 | 0.132 | 13.6 | 6.38 |
| | 280 | 0.351 | 0.224 | 14.82 | 10.05 | 0.360 | 0.227 | 14.4 | 9.70 |
| | 300 | 0.473 | 0.091 | 20.54 | 3.72 | 0.477 | 0.068 | 21.0 | 2.99 |
| | 320 | 0.616 | 0.143 | 26.01 | 6.36 | 0.616 | 0.143 | 25.7 | 6.15 |
| | 340 | 0.770 | 0.179 | 33.49 | 8.26 | 0.769 | 0.177 | 32.1 | 6.94 |
| | 360 | 0.409 | 0.309 | 16.50 | 14.00 | 0.400 | 0.289 | 16.2 | 13.9 |
| | 380 | 0.691 | 0.177 | 29.64 | 7.90 | 0.690 | 0.166 | 29.1 | 9.49 |
| | >400 | 0.741 | 0.484 | 32.00 | 22.24 | 0.812 | 0.530 | 31.3 | 24.6 |

The volume and value for the sample trees by main product class after merchandizing using the optimum bucking method for the CTL and TL specifications are summed in Table 3. For both regions, the volume recovery for pulpwood favored TL merchandizing, and sawtimber merchandizing favored CTL. The recovery of chip-n-saw was not consistent

between regions. The value recovery incorporated the volume recovery and the small differences in value between the regions. The total value recovery was similar across region and system, with a total range of $13.12 or less than $1.00 per cubic meter. The TL merchandizing recovered a greater total volume, but CTL merchandizing recovered a greater value. Observing the value recovery by diameter class, the DSH classes of up to 100 mm had a similar value for both merchandizing systems, but CTL merchandizing yielded a greater or similar value per cubic meter across the range in the larger tree sizes (Figure 3).

**Table 3.** Total volume and value recovery by product class using optimum bucking method per region and system.

| Region | Product | Cut-to-Length | | Tree-Length | |
|---|---|---|---|---|---|
| | | Volume (m³) | Value ($) | Volume (m³) | Value ($) |
| Piedmont | Pulpwood | 1.33 | 43.01 | 2.74 | 88.90 |
| | Chip-n-saw | 3.87 | 140.73 | 4.77 | 173.38 |
| | Sawtimber | 11.97 | 526.75 | 9.91 | 436.13 |
| | Total | 17.17 | 710.50 | 17.43 | 698.41 |
| Coastal Plain | Pulpwood | 0.62 | 20.10 | 3.10 | 100.32 |
| | Chip-n-saw | 5.05 | 183.39 | 4.67 | 169.35 |
| | Sawtimber | 11.52 | 508.04 | 9.77 | 429.95 |
| | Total | 17.18 | 711.53 | 17.54 | 699.62 |

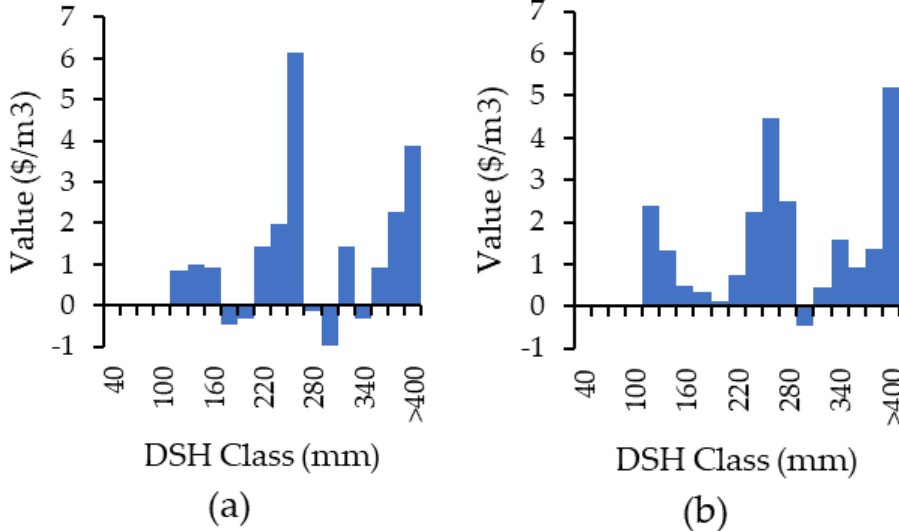

**Figure 3.** Difference in value recovery total between CTL and TL systems using optimal bucking for the Piedmont (**a**) and Coastal Plain (**b**).

## 4. Discussion

We captured the tree data from a harvester, where the operator made bucking decisions using visual quality indicators (scars, swells, large branch diameters, crooks, and sweeps) and the assistance of the on-board computer and the specifications entered into it. Relatively few of the trees were merchandized at or beyond the SED limits for pulpwood specifications. The amounts of pulpwood recovery and waste were probably underestimated as a result. Small, although sometimes significant, differences were observed between the volume and value recovery for the TL and processor-based merchandizing systems [14,17,18,36]. Operations producing out-of-specification logs affect comparisons between modeled and real results [17,18]. A combination of the tree geometry, mill specifications, and prices would affect the comparison between CTL and TL outcomes [14].

Merchandizing studies in the southern USA have used a variety of tree data, ref. [37] including tree diameter and height, quality indicators, and a taper equation to merchandize according to a number of strategies. Other studies have relied on specific sample tree measurements as the data for optimization or comparison [10,18,36,38]. Finally, other comparisons have been accomplished by merchandizing similar areas or groups of trees using different systems [14,17]. The results of these studies might be generalizable based on similar tree dimensions, but market and price differences could limit their applicability to other regions or at other times.

Accumulating tree data would rely on a mechanized processing or harvesting head or manual measurements of felled trees or scanning felled or standing trees [39–41]. Each of the sources of tree data presents a barrier to a landowner or timber buyer either due to its scarcity, labor intensity, or technical complexity. Additionally, some of the visual indicators of stem quality (e.g., cankers, scars, large limbs) are important in bucking decisions but not obvious in stem profile data. Tree data from regional utilization studies [42] may provide both stem profile data and quality indicators. For optimization, the creation of stem profiles would require the interpolation of stem diameters between measurement locations, typically using standard log lengths.

Optimization routines are included in the software of harvester and processor head manufacturers. The sensors collect taper data from recently processed trees and can use these relationships to predict the optimal solution based on the stump diameter. We used an optimization technique that could be available to anyone using an Excel spreadsheet and Visual Basic. The method required technical expertise since all of the possible bucking lengths for each tree had to be constructed. Several researchers have developed optimization programs that can utilize tree taper and curvature data [41,43–45]. Depending on the funding sources used to develop and validate these models, public availability might be limited.

## 5. Conclusions

We chose to focus on gross value since it simplified the analysis and reduced the number of assumptions made. We acknowledge that individual wood buyers or loggers would need to estimate the net value to complete the analysis. With a limited number of trees across two markets, we found that the gross value recovery slightly favored CTL merchandizing. The volume recovered was higher for TL merchandizing across most of the tree size classes studied. The limited variety in log lengths available for CTL merchandizing likely played a role in that result. Conrad et al. emphasized that the current mill policies in the South have discouraged adoption of CTL systems, and the lack of manufacturers and dealers of CTL systems in the region further disincentivize the adoption of the systems [4]. Diniz et al. confirmed the limited use of CTL systems, including harvesters or processors, across the southeastern U.S.A [46]. Many of the same barriers (e.g., harvesting costs, capital investment) to CTL adoption noted by Gellersted and Dahlin may still be important [9].

Market participants need access to better tools to evaluate merchandizing alternatives since both markets and prices are dynamic. Recently, both widespread investment in sawmills and the continued loss of regional pulp and paper capacity highlight the need for flexible merchandizing alternatives and quantitative tools for evaluating the options.

**Author Contributions:** Conceptualization, C.D., M.S. and J.S.; methodology, C.D., M.S. and J.S.; software, C.D. and J.S.; validation, C.D. and M.S.; formal analysis, C.D.; writing—original draft, C.D.; writing—review & editing, M.S. and J.S.; visualization, C.D., M.S. and J.S.; supervision, M.S. and J.S.; project administration, M.S.; funding acquisition, M.S. All authors have read and agreed to the published version of the manuscript.

**Funding:** This research was supported in part by an appointment to the United States Forest Service (USFS) Research Participation Program, administered by the Oak Ridge Institute for Science and Education (ORISE) through an interagency agreement between the U.S. Department of Energy (DOE) and the U.S. Department of Agriculture (USDA). ORISE is managed by ORAU under DOE contract number DE-SC0014664. This paper was written and prepared by a U.S. Government employee on official time, and therefore it is in the public domain and not subject to copyright. This research was supported by the USDA Forest Service. The findings and conclusions in this publication are those of the author(s) and should not be construed to represent an official USDA, Forest Service, or U.S. Government determination or policy.

**Data Availability Statement:** Data are contained within the article.

**Acknowledgments:** The authors would like to acknowledge Preston Green and Miller Timber for the tree data, Simon Ackerman for assistance with the *.stm files, FP Innovations for making the StanForD2Tbl software available, Edd Watson for assistance with the product specification data, and the mill personnel in South Carolina and Georgia for providing the product specifications.

**Conflicts of Interest:** The authors declare no conflicts of interest.

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
