# Peer review of "Applying an Optimum Bucking Method to Comparing the Volume and Value Recovery of Cut-to-Length and Tree-Length Merchandizing Systems in Piedmont and the Coastal Plain"

_forests, doi:10.3390/f15030550_

Round 1

Reviewer 1 Report

Comments and Suggestions for Authors

I appreciate the authors’ effort to writing the research paper. I have reviewed the paper “Applying optimum bucking method to compare volume and value recovery of cut-to-length and tree-length merchandizing systems in Piedmont and Coastal Plain”. I think this paper could be accepted.

I suspect it will be species-specific, but you'll need to explain how that can be addressed and what species it's suitable for.  This requires explanation.

Comments on the Quality of English Language

I appreciate the authors’ effort to writing the research paper. I have reviewed the paper “Applying optimum bucking method to compare volume and value recovery of cut-to-length and tree-length merchandizing systems in Piedmont and Coastal Plain”. I think this paper could be accepted.

Reviewer 2 Report

Comments and Suggestions for Authors

Manuscript presents an interesting research topic and brings useful information suggesting solutions for selection of optimum bucking method in timber harvesting.

Minor revision following listed comments is suggested:

L63-L65: Change of citation style should be considered (e.g. „[28] developed an optimum bucking solution called BUCK.“ To: „Sessions et al. [28] developed an optimum bucking solution called BUCK.“ or „Sessions et al. developed an optimum bucking solution called BUCK [28].“

L91: Tree species should be stated.

Please consider adding a sentence supported by citations, either in Discussion or Conclusions, regarding the (different) cost of harvesting between CTL and TL harvesting systems.

Reviewer 3 Report

Comments and Suggestions for Authors

General:

The topic is relevant to Forests readers, but the authors need to acknowledge the International audience. They make too many assumptions about the region and subject matter, thereby losing the reader. For instance, neither the abstract nor introduction mention whether the trees/forests are softwoods or hardwoods. Loblolly pine is mentioned in key words, but for readers not familiar with the plantation forestry in the SE US, they need a more explicit statement.

The first half of the introduction is difficult to follow. I am not sure what is relevant and what is not. It does not set up the study objectives. It is not until the conclusions that the reader is informed that CTL adoption in the SE US is lagging. Is your study rationale that CTL adoption will lead to higher value recovery? Or is your study rationale more about bucking optimization techniques using Excel and Visual Basic?

It is still not clear to me how the study was conducted. As I understand it, the CTL component was done in the woods by the harvester. What is not clear is how the TL component was done. Were tree lengths taken to mill and bucked up in the logyard? Was it a simulation? It is critical that you walk the reader through this.

Overall, the paper does not flow well. The writing is very choppy and needs editing.

Specific comments:

Line 13 – compared to what?

Line 22 – 13 billion ft3 when?

Lines 23-25 – what is the relevance of this?

Line 27 – what is the context of 37,000 tons?

Line 31 – what factors are you referring to?

Line 34 – what is the relevance of the years 1979-1987? Is the introduction chronological?

Line 35 – provide an example

Line 38 – when?

Line 45 – define “processed based merchandising”

Line 50 – clarify that sawmills are doing the bucking in the logyard? Why can TL lead to errors? Provide examples.

Line 54 – by traditional bucking you mean with a chainsaw, correct? This is still common with high quality hardwoods in which the stakes are much higher. Obviously, not practical in a loblolly pine plantation.

Line 101 – 57 trees from each region? Please define the stump diameter classes. Is it 40 – 320? Figure 1 uses milimeters (mm). Is that a typo?

Line 107 – does figure 1 represent 4957 stems? Is 20mm class a typo? Shouldn’t it be cm.

Line 187 - Table 2 also uses milimeters. Can you have a stump diameter of 40 mm? You must mean 40 cm. In Table 2 is N=57 for each region? For each DSH class, three stems are represented?

Line 202 – why not add total values for CTL and TL so reader can compare?

Line 205 – is this figure relevant? Do we really care about the difference between the two regions?

Comments on the Quality of English Language

needs editing

Reviewer 4 Report

Comments and Suggestions for Authors

Dear Authors,

I have reviewed the paper titled: “Applying optimum bucking method to compare volume and value recovery of cut-to-length and tree-length merchandizing systems in Piedmont and Coastal Plain". In my opinion, the aims of the paper are germane with “Forests” journal topic and SI topic, however in the present form, the paper has some flaws. The contribution of this paper to the scientific knowledge is average. I understand the difficult work done, but as a reviewer it is my duty to highlight the gaps in order to improve the research approach and its presentation to the international scientific community. In general, the paper is well written and you managed to clearly explain the rationale of the research. However, I think that in the present form the novelty of the research is not enough to achieve publication in a Q1 journal. Considering this evaluation, I attach a file with detailed comments and suggestions in order to improve this paper.

Round 2

Reviewer 3 Report

Comments and Suggestions for Authors

The introduction remains confusing, particularly to an international readership.

The authors remain provincial and do not provide implications beyond the study region.

Comments on the Quality of English Language

Still needs editing.

Reviewer 4 Report

Comments and Suggestions for Authors

Dear authors,

I congratulate you on the work done, even if your comments on my suggestions on the discussion and conclusion chapter do not resolve the problems I had highlighted. However, these are formal problems that would only have increased the form of the scientific paper. I therefore believe that the document has reached the standards for a scientific publication while maintaining those small imperfections.